# Intra- and Interobserver Reliability Comparison of Clinical Gait Analysis Data between Two Gait Laboratories

**René Schwesig** [1,*], **Regina Wegener** [2], **Christof Hurschler** [3], **Kevin Laudner** [4] **and Frank Seehaus** [5]

1   Department of Orthopaedic and Trauma Surgery, Martin-Luther-University Halle-Wittenberg, Ernst-Grube-Str. 40, 06120 Halle, Germany
2   Private Practice for Neurology Dr. Wegener, Lienaustr. 2, 23730 Neustadt, Holstein, Germany; reginawegener@gmx.net
3   Laboratory for Biomechanics and Biomaterials, Hannover Medical School, Anna-von-Borries-Str. 1-7, 30625 Hannover, Germany; hurschler.christof@mh-hannover.de
4   Department of Health Sciences, University of Colorado, Colorado Springs, CO 80918, USA; klaudner@uccs.edu
5   Department of Orthopaedic Surgery, Friedrich-Alexander-University Erlangen-Nürnberg, Rathsberger Str. 57, 91054 Erlangen, Germany; frank.seehaus@fau.de
*   Correspondence: rene.schwesig@uk-halle.de; Tel.: +49-345-557-1317; Fax: +49-345-557-4899



**Featured Application: Clinical gait analysis (CGA) is an in vivo method used to measure the movement behavior/gait patterns of patients before and after an orthopedic treatment. It monitors joint kinematics and kinetics under dynamic conditions within the musculoskeletal system. This study contributes to a better understanding of the comparability and significance of motion analysis data recorded in different gait laboratories using different technical qualities.**

**Abstract:** Comparing clinical gait analysis (CGA) data between clinical centers is critical in the treatment and rehabilitation progress. However, CGA protocols and system configurations, as well as choice of marker sets and individual variability during marker attachment, may affect the comparability of data. The aim of this study was to evaluate reliability of CGA data collected between two gait analysis laboratories. Three healthy subjects underwent a standardized CGA protocol at two separate centers. Kinematic data were captured using the same motion capturing systems (two systems, same manufacturer, but different analysis software and camera configurations). The CGA data were analyzed by the same two observers for both centers. Interobserver reliability was calculated using single measure intraclass correlation coefficients (ICC). Intraobserver as well as between-laboratory intraobserver reliability were assessed using an average measure ICC. Interobserver reliability for all joints (ICC$_{total}$ = 0.79) was found to be significantly lower ($p < 0.001$) than intraobserver reliability (ICC$_{total}$ = 0.93), but significantly higher ($p < 0.001$) than between-laboratory intraobserver reliability (ICC$_{total}$ = 0.55). Data comparison between both centers revealed significant differences for 39% of investigated parameters. Different hardware and software configurations impact CGA data and influence between-laboratory comparisons. Furthermore, lower intra- and interobserver reliability were found for ankle kinematics in comparison to the hip and knee, particularly for interobserver reliability.

**Keywords:** motion analysis; kinematics, repeatability; lower extremity; optical infrared camera motion capturing system

## 1. Introduction

Three-dimensional clinical gait analysis (CGA) is an important diagnostic tool within the field of movement disorders [1]. Optical marker tracking systems are considered the gold standard for this type of assessment [2,3]. However, discrepancies in CGA data across multiple analyses can be caused by differences in marker sets [4], observer error [5], marker placement [6] and patient variability [7]. Despite applied standardized protocols (e.g., marker-set, measurement pipeline), incompatibilities and discrepancies within CGA data of the same patient cohorts collected by different institutions are often observed, and have caused reservations concerning the applicability of CGA among multicenter studies [3,4,8]. Ferrari et al. [4] compared the trunk, pelvis and lower limb kinematics of five separate measurement protocols (different marker sets), using a single data pool of subjects. These authors reported good correlations between kinematic variables in the sagittal plane (flexion/extension), but poorer correlations for out-of-sagittal-plane rotations, such as knee abduction/adduction. They also reported good correlations among protocols with similar biomechanical models [4]. Ferrari et al. [4] concluded that the comparison of different measurement protocols results in higher data variability when compared to interobserver and interlaboratory comparisons for most gait characteristics. Obviously, model conventions and definitions are crucial for data comparison. Gorton et al. [8] identified the marker placement procedure of examiners as the largest source of error. The use of a standardized protocol for marker placement decreased data variability by up to 20%. Based on clinical experience, the authors of the current study hypothesized that significant discrepancies would exist between CGA data sets collected using different hardware and software infrastructures or configurations.

Due to the observed limitations in the reproducibility of CGA data, efforts have focused on the reduction of measurement error, with the aim of producing data with the highest possible degree of reproducibility. To improve our understanding of the accuracy and reproducibility of CGA data, cross-laboratory studies must be performed using a single cohort of subjects. Previous research has reported on the repeatability of motion capture data using separate trials, sessions and observers under various conditions for a specific CGA setup or laboratory [5–7,9,10]. However, reports for multi-center repeatability are limited and have focused on differences in hardware configurations, marker placement, as well as between trials and days of measurement [3,8,11]. Unfortunately, a validation using the same motion capturing technology, but different hardware and software infrastructure/configurations (e.g., capturing software, camera type), among a single cohort of subjects is lacking.

It is critical for clinicians and researchers to have reliable examination tools to accurately and objectively assess the functional status of a joint [12–16]. A high level of intraobserver reliability is imperative to accurately evaluate the longitudinal effects of the rehabilitation process and to identify differences between subjects [17,18]. A central research question is whether the findings of clinical gait analyses conducted by multiple laboratories are consistent and reliable enough for making clinical decisions—or is the dependence on observers, repeat measurements or measurement and analysis protocols too large?

The specific aims of this study were to evaluate the intra- and interobserver reliability and the equivalence of CGA data between two gait analysis laboratories (between-laboratory intraobserver reliability) using the same motion capturing technology, but different hardware and software infrastructure/configurations among a single cohort of subjects. The following two hypotheses were specifically investigated:

1) Intraobserver reliability of CGA data for the lower extremity obtained for the same cohort of subjects and captured at one laboratory will be good-to-excellent, whereas interobserver reliability will be fair-to-good.
2) CGA data for the lower extremity obtained for the same cohort of subjects, captured at two separate laboratories by the same observers, using different hardware and software systems, will be equivalent.

## 2. Materials and Methods

### 2.1. Subjects

A standardized measurement protocol was performed repeatedly on three adult subjects (one female, two males) who were asymptomatic at the time of testing and did not display any movement disorders. This cohort had a mean (±SD) age of 33.3 (±4.0) years, mean body weight of 77.3 (±16.3) kg, mean body height of 176.7 (±9.1) cm, and a mean body mass index of 24.3 (±2.9) kg·m$^{-2}$ (Table A1). The study was approved by the ethical committee of Martin-Luther-University Halle-Wittenberg (approval number: 217/08.03.10/10). Written consent from study participants was obtained prior to data collection.

### 2.2. Measurement Set-Up

Gait analyses were performed in two gait laboratories (Figure 1). Both laboratories used an optical infrared camera-based motion capturing system provided by one manufacturer (Vicon Motion System Ltd., Oxford, UK).

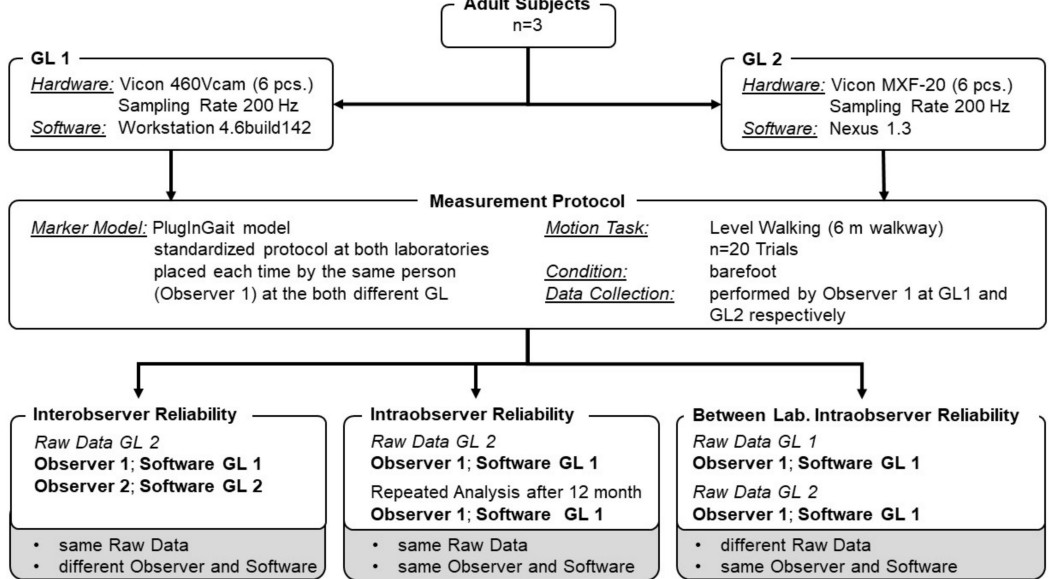

**Figure 1.** Data processing chart—clinical gait analysis (CGA) data collected on three healthy test-subjects (*n* = 20 barefoot trials per person). The data were captured using the same measurement protocols in both the first (GL1) and the second laboratories (GL2). Three reliability tests were performed with the datasets from GL1 and GL2. Interobserver reliability was tested for two observers using the CGA data collected at GL2. Observer 1 used the data processing software from GL 1 (Workstation 4.6), which was different to that of Observer 2 (GL2: Nexus 1.3). To assess intraobserver reliability, the same data set was analyzed a second time by observer 1 after a time interval of 12 months, using the CGA data collected at GL2 and the software from GL1. To test between-laboratory intraobserver reliability, the CGA data collected at both sites (GL1 and GL2) were analyzed by Observer 1 using software from GL1.

The first laboratory (GL1) was equipped with a six-camera system (460 Vcam cameras, Workstation 4.6 build 142 software). The second laboratory (GL2) used a six-camera system (MXF-20 cameras, Nexus 1.3 software). These systems thus differed in their respective configurations, both in terms of analysis software (Workstation vs. Nexus) and camera type (Vcam vs. MXF). Both laboratories applied the same sampling rate (200 Hz) and the same reflective markers (14 mm diameter). Capture space represented the full dimensions (length, width and height) of the motion capture room for GL1 (7 m, 4 m and 3 m) and for GL2 (15.5 m, 8.8 m and 4.8 m). Capture volume was defined as the area within the capture space, for which motion capturing cameras were able to capture the motion task of each

subject. The GL1 capture volume was 6 m, 2 m and 2 m. The capture volume for GL2 was 10.0 m, 8.5 m and 2.9 m. Captured marker data were processed and trajectories were labeled using the PlugInGait model under a standardized protocol at both laboratories. All kinematic data were Woltring filtered using a mean squared error setting of 10. These gait data were reduced to 100% of one gait cycle using gait cycle event detection, based on the available force plates (threshold: 20 N). All subsequent measurement conditions were consistent at both laboratories.

### 2.3. Measurement Protocol

Calibration of the optical infrared camera-based motion capturing systems at both laboratories were performed according to the manufacturer's guidelines. This calibration process consisted of two main steps: (*i*) a static calibration to calculate the origin of the capture volume and define the 3D workspace orientation (x, y, z directions) and (*ii*) a dynamic calibration to calculate the relative position and orientation of each camera within the capture volume. Calibration quality was checked according to the manufacturer's guidelines.

The PlugInGait marker setup for the lower extremity (kinematic model V 2.3) was used in both laboratories, based on the work of Kadaba et al. [19]. Markers were attached to each subject by the same experienced staff member (Observer 1) at both gait analysis laboratories, according to a standardized protocol for anthropometric measurements, landmark identification and marker mounting.

Each of the three subjects performed 20 barefoot gait trials with a self-selected walking speed resulting in a total of 60 CGA trials per gait laboratory. Individual gait speed was controlled and standardized between data collection sessions. The same two staff members (Observer 1 and Observer 2) performed data processing and analysis at both laboratories, using a standardized protocol for data processing, labeling and gait event detection. For each reliability analysis variation, the observer started with the original raw data in their data processing routine. These raw data were labeled, gait events were detected and possible gaps of reconstructed marker trajectories were filled. The kinematic parameters were extracted using the same template and again used a standardized workflow for both observers. Kinematic data used for reliability analysis, which consisted of specific movement parameters in the sagittal and frontal plane for the hip, knee and ankle. Parameters were selected according to the specifications of Benedetti et al. [20] (Figure 2).

Data from the right leg of each subject were used for the analyses of reliability. Three different measures of reliability were performed (Figure 1):

- Interobserver Reliability: The reliability of the two observers, using the same data set, was assessed using separate analysis software (Observer 1: GL1 (Workstation); Observer 2: GL2 (Nexus)).
- Intraobserver Reliability: The reliability of the same data set among the same observer was tested. This assessment was performed with a time interval of 12 months between analyses using the same software (Workstation). The CGA data for this assessment were collected at the GL2 site.
- Between-Laboratory Intraobserver Reliability: To compare the effect of laboratory environment and instrumentation while excluding observer-dependent influences, CGA data collected at both laboratories were analyzed by a single observer using the same analysis software.

The following reliability variables were assessed. $ICC_{mean}$ was the average of all parameters for hip, knee and ankle. $ICC_{total}$ (Equation (1)) was the average of hip $ICC_{mean}$, knee $ICC_{mean}$ and ankle $ICC_{mean}$ for all three types of reliability (intraobserver, interobserver, between-laboratory intraobserver).

$$ICC_{total} = hip\ ICC_{mean} + knee\ ICC_{mean} + ankle\ ICC_{mean}\ /\ 3 \qquad (1)$$

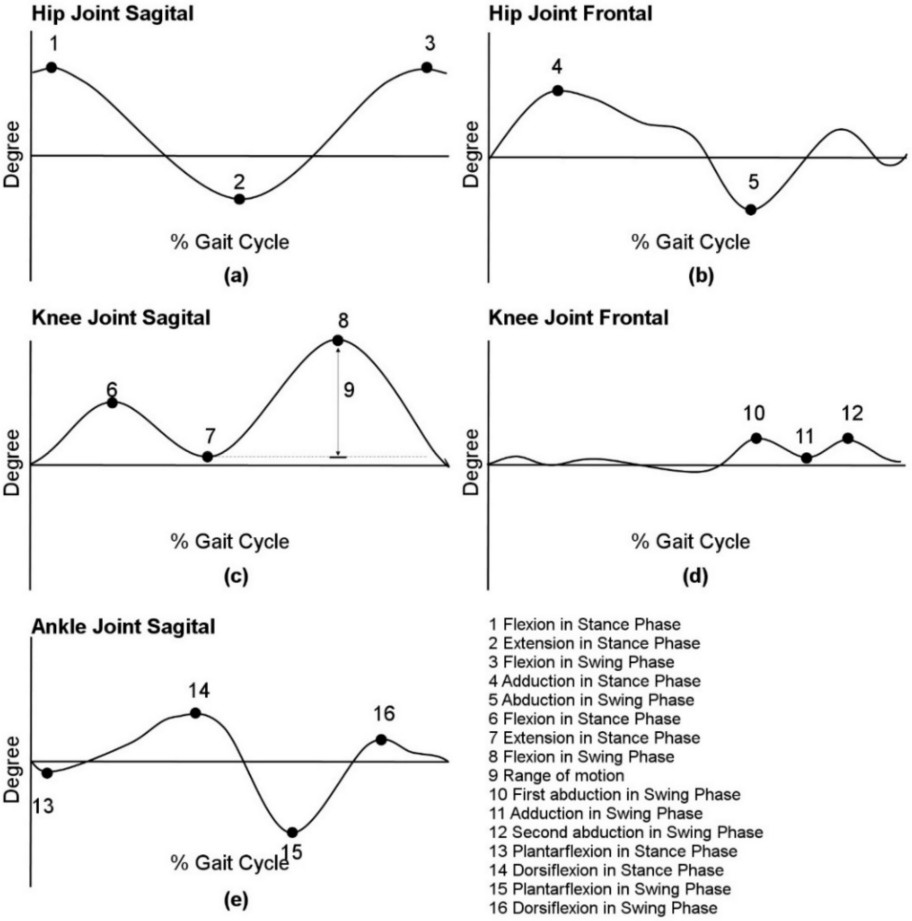

**Figure 2.** Parameters extracted (No. 1 to 16) out of processed kinematic data for sagittal (**a**) and frontal (**b**) plane of the hip joint, for sagittal (**c**) and frontal (**d**) plane of the knee joint, as well as data for sagittal (**e**) plane of the ankle joint.

## 2.4. Statistics

Descriptive statistics (mean, standard deviation) were based on 20 barefoot trials and were calculated for 31 kinematic parameters in the sagittal and frontal plane for the hip, knee and ankle joints. Reliability analyses were divided into five parts:

1.  For interobserver reliability, a single measure intraclass correlation coefficient (ICC) was calculated. The number of measures (k) was 60 (*n* = 20 barefoot trials from three subjects).
2.  For intraobserver reliability, an average measure ICC was calculated. The number of measures (k) was 60.
3.  To assess between-laboratory intraobserver reliability, an average measure ICC was calculated and again referenced to the same ICC value classification [21]. The ICC indicated excellent reliability if the value was above 0.75, fair-to-good reliability between 0.40 and 0.75 and poor reliability when less than 0.40. A two-way mixed-effects model (definition: absolute agreement) was used for all calculations. For all ICC values, 95% confidence intervals were reported.
4.  To estimate experimental errors of a joint angle, the standard error was calculated (for intra- ($\sigma^{repeated}$), inter- ($\sigma^{observer}$) and between-laboratory intraobserver reliability ($\sigma^{sess(lab)}$)) as described by Schwartz et al. [10], as well as the magnitude of the interobserver error and its ratio to intra-subject error r (Equations (2)–(4)).

$$r = \sigma^{repeated}/\sigma^{trial} \tag{2}$$

$$r = \sigma^{observer}/\sigma^{trial} \tag{3}$$

$$r = \sigma^{sesss(lab)}/\sigma^{trial} \tag{4}$$

5.  A scatter-plot technique suggested by Bland and Altman [22] was used to assess interchangeability (equivalence) of CGA data between laboratories. Calculated differences for joint angles were plotted against their average value for each subject. The interchangeability of CGA data was tested by a bounding criterion defined as the mean ± two standard deviations of the measured differences (approximately 95% of all measured values).

6.  To evaluate the variability within and between subjects, the standard error of measurement (SEM) was calculated in conjunction with the ICCs, using the following equation from Portney and Watkins [23]:

$$SEM = SD \times \sqrt{(1 - ICC)} \tag{5}$$

    ICC values may be influenced by inter-subject variability of scores, because a large ICC may be reported despite poor trial-to-trial consistency if the inter-subject variability is too high [23,24]. However, the SEM is not affected by inter-subject variability [24].

7.  Mean differences for multicenter comparisons were tested using variance analysis. A one-factor, univariate general linear model (GLM; dependent variable: hip flexion during stance; independent variable: CGA center; covariate: walking speed) was performed. Prior to inference statistical analyses, all variables were tested for normal distribution (Kolmogorov–Smirnov test). To estimate practical relevance and to quantify the differences between GL1 and GL2, effect sizes (partial eta squared, $\eta_p^2$ [25,26]) were calculated for the main effects ($\eta_p^2$) and the mean differences divided by the pooled standard deviations (d). To evaluate effect sizes, d or $\eta_p^2$ were classified, with $d \geq 0.2$, $d \geq 0.5$, $d \geq 0.8$ or $\eta_p^2 \geq 0.01$, $\eta_p^2 \geq 0.06$, $\eta_p^2 \geq 0.14$ indicating small, medium or large effects, respectively [27]. Level of significance was adjusted to $p \leq 0.002$ (0.05/31 = 0.002) for multiple tests by means of a Bonferroni correction.

All statistical analyses were performed using SPSS version 25.0 for Windows (SPSS Inc., IBM, Armonk, NY, USA).

## 3. Results

Because of the controlled and standardized CGA data collection, walking speed and stride length parameters did not differ between laboratory measurement sessions (walking speed: $1.40 \pm 0.06$ m·s$^{-1}$ vs. $1.40 \pm 0.04$ m·s$^{-1}$; stride length: $1.38 \pm 0.09$ m vs. $1.40 \pm 0.10$ m).

### 3.1. Interobserver Reliability

All interobserver reliability variables fulfilled the assumption of normality. Interobserver reliability at one laboratory (GL2), showed excellent reliability for 71% of the parameters observed (22/31). Based on the lower limit of the 95% CI, 58% (18/31) displayed an ICC value of at least 0.75. Nineteen percent (6/31) of the parameters showed fair-to-good reliability, whereas 10% (3/31) were poor (Table 1).

**Table 1.** Single site inter- and intraobserver intraclass correlation coefficient (ICC) values and results of repeated single observer analysis at both laboratories (between-laboratory intraobserver values) of an ICC greater than 0.75 were considered excellent and marked in bold.

| Parameter | Intraobserver | | | Interobserver | | | Between-Laboratory Intraobserver | | |
|---|---|---|---|---|---|---|---|---|---|
| | ICC | 95% CI | | ICC | 95% CI | | ICC | 95% CI | |
| | | lower | upper | | lower | upper | | lower | upper |
| **Hip (degree)** | | | | | | | | | |
| *Flexion ST* | **1.00** | **1.00** | 1.00 | **0.97** | **0.95** | 0.98 | 0.60 | 0.10 | 0.80 |
| *Extension ST* | **1.00** | **1.00** | 1.00 | **0.76** | 0.63 | 0.85 | 0.72 | 0.54 | 0.83 |
| *Flexion SW* | **1.00** | **1.00** | 1.00 | **0.99** | **0.97** | 0.99 | 0.71 | 0 | 0.89 |
| *Adduction ST* | **1.00** | **1.00** | 1.00 | **0.98** | **0.97** | 0.99 | **0.88** | 0.55 | 0.95 |
| *Abduction SW* | **1.00** | **1.00** | 1.00 | **0.93** | **0.89** | 0.96 | 0.59 | 0.06 | 0.80 |
| **Hip (% gait cycle)** | | | | | | | | | |
| *Flexion ST* | **1.00** | **0.99** | 1.00 | **0.96** | **0.92** | 0.97 | **0.90** | 0.83 | 0.94 |
| *Extension ST* | **0.92** | **0.77** | 0.97 | **0.81** | 0.64 | 0.90 | 0.46 | 0.00 | 0.71 |
| *Flexion SW* | **0.83** | 0.72 | 0.90 | 0.71 | 0.56 | 0.81 | 0.31 | 0.00 | 0.57 |
| *Adduction ST* | **1.00** | **1.00** | 1.00 | **0.95** | **0.92** | 0.97 | **0.78** | 0.63 | 0.87 |
| *Abduction SW* | **0.99** | **0.98** | 0.99 | **0.94** | **0.90** | 0.97 | 0.53 | 0.23 | 0.72 |
| **Knee (degree)** | | | | | | | | | |
| *Flexion ST* | **1.00** | **1.00** | 1.00 | **0.91** | **0.86** | 0.95 | 0.57 | 0.08 | 0.78 |
| *Extension ST* | **1.00** | **1.00** | 1.00 | **1.00** | **0.99** | 1.00 | 0.74 | 0.56 | 0.84 |
| *Flexion SW* | **1.00** | **1.00** | 1.00 | 0.47 | 0.25 | 0.65 | **0.82** | 0.70 | 0.89 |
| *Range of motion* | **1.00** | **1.00** | 1.00 | 0.40 | 0.17 | 0.59 | 0.46 | 0.10 | 0.68 |
| *First abduction SW* | **1.00** | **1.00** | 1.00 | **0.99** | **0.98** | 0.99 | 0.00 | 0.00 | 0.35 |
| *Adduction SW* | **1.00** | **1.00** | 1.00 | **0.97** | **0.94** | 0.98 | 0.41 | 0.00 | 0.67 |
| *Second abduction SW* | **1.00** | **1.00** | 1.00 | **0.97** | **0.95** | 0.96 | 0.41 | 0.00 | 0.67 |
| **Knee (% gait cycle)** | | | | | | | | | |
| *Flexion ST* | **0.99** | **0.98** | 0.99 | **0.88** | **0.81** | 0.93 | **0.81** | 0.18 | 0.93 |
| *Extension ST* | **0.98** | **0.97** | 0.99 | **0.94** | **0.83** | 0.97 | 0.21 | 0.00 | 0.53 |
| *Flexion SW* | **0.89** | 0.66 | 0.95 | **0.84** | **0.75** | 0.90 | 0.13 | 0.00 | 0.42 |
| *First abduction SW* | **0.97** | **0.93** | 0.98 | **0.84** | 0.74 | 0.90 | 0.48 | 0.12 | 0.69 |
| *Adduction SW* | **0.98** | **0.92** | 0.99 | 0.72 | 0.55 | 0.83 | 0.19 | 0.00 | 0.52 |
| *Second abduction SW* | **0.98** | **0.81** | 1.00 | **0.91** | **0.85** | 0.95 | **0.75** | 0.55 | 0.86 |
| **Ankle (degree)** | | | | | | | | | |
| *Plantarflexion ST* | 0.23 | 0.00 | 0.51 | 0.11 | 0 | 0.314 | 0.33 | 0.00 | 0.60 |
| *Dorsiflexion ST* | 0.61 | 0.28 | 0.78 | 0.47 | 0.02 | 0.721 | 0.74 | 0.57 | 0.85 |
| *Plantarflexion SW* | **0.98** | **0.94** | 0.99 | **0.93** | **0.79** | 0.968 | **0.86** | 0.62 | 0.94 |
| *Dorsiflexion SW* | **0.82** | 0.63 | 0.90 | 0.72 | 0.25 | 0.875 | **0.83** | 0.65 | 0.91 |
| **Ankle (% gait cycle)** | | | | | | | | | |
| *Plantarflexion ST* | **0.98** | **0.96** | 0.99 | **0.82** | 0.69 | 0.90 | **0.77** | 0.07 | 0.92 |
| *Dorsiflexion ST* | **0.95** | **0.92** | 0.97 | 0.22 | 0.00 | 0.44 | 0.36 | 0.00 | 0.62 |
| *Plantarflexion SW* | **0.92** | **0.82** | 0.96 | **0.84** | **0.75** | 0.90 | 0.27 | 0.00 | 0.55 |
| *Dorsiflexion SW* | **0.95** | **0.92** | 0.97 | 0.50 | 0.26 | 0.68 | 0.45 | 0.06 | 0.68 |
| **ICC ≥ 0.75 (%)** | **94% (29/31)** | | | **71% (22/31)** | | | **29% (9/31)** | | |

**Remarks:** ST = stance phase; SW = swing phase.

The highest ICC value was observed for knee extension in the stance phase (ICC = 0.97, excellent reliability). In contrast, dorsiflexion during the stance phase showed poor reliability (ICC = 0.22) (Table 1). After averaging all parameters for interobserver reliability, an $ICC_{total} = 0.79$ (95% CI: 0.67–0.86) was observed (Table 2).

For interobserver reliability, the largest standard error was observed for hip adduction (% gait cycle) during stance phase, as well as for ankle dorsiflexion (% gait cycle) during swing phase, with each presenting with a standard error of $\sigma^{observer} = 4.4\%$ (Table 3). Plantarflexion during the stance phase had the worst detected angle ($\sigma^{observer} = 3.8°$). The largest ratio of interobserver to intrasubject error was observed for knee abduction angle in the stance phase ($\sigma^{observer} = 2.4°$, r = 3.2; Table 3).

**Table 2.** Calculated mean ICC values (ICC$_{mean}$ = mean of all parameters for hip, knee and ankle) and 95% confidence intervals (95% CI) for the ankle, knee and hip, as well as total mean ICC values (ICC$_{total}$ = ICC$_{mean}$ hip + ICC$_{mean}$ knee + ICC$_{mean}$ ankle). Values for ICC above 0.75 were considered excellent and marked in bold.

| Joint | Intraobserver (95% CI) | Interobserver (95% CI) | Between-Laboratory Intraobserver (95% CI) |
|---|---|---|---|
| Hip ∅ ICC$_{mean}$ | **0.97 (0.95–0.99)** | **0.90 (0.84–0.94)** | 0.65 (0.27–0.81) |
| Knee ∅ ICC$_{mean}$ | **0.98 (0.94–0.99)** | **0.83 (0.74–0.89)** | 0.46 (0.05–0.68) |
| Ankle ∅ ICC$_{mean}$ | **0.81 (0.66–0.88)** | 0.58 (0.33–0.73) | 0.58 (0.21–0.76) |
| ∅ **ICC$_{total}$** | **0.93 (0.87–0.96)** | **0.79 (0.67–0.86)** | 0.56 (0.16–0.74) |
| **Analysis of Variance (ICC$_{total}$)** | Comparison of all three types of reliability: $p < 0.001$; $\eta_p^2 = 0.533$ Intraobserver vs. Interobserver: $p < 0.001$; $\eta_p^2 = 0.399$ Intraobserver vs. Between-laboratory intraobserver: $p < 0.001$; $\eta_p^2 = 0.676$ Interobserver vs. Between-laboratory intraobserver: $p < 0.001$; $\eta_p^2 = 0.374$ | | |

**Remarks:** ICC$_{mean}$ = mean of all parameters for hip, knee and ankle; ICC$_{total}$ = ICC$_{mean}$ hip + ICC$_{mean}$ knee + ICC$_{mean}$ ankle divided by 3 (for all three types of reliability).

**Table 3.** Standard errors for intra- ($\sigma^{repeated}$), inter- ($\sigma^{observer}$) and between-laboratory intraobserver ($\sigma^{sess(lab)}$) reliability. The inter-trial error is represented by r (r = $\sigma^{repeated}$ / $\sigma^{trial}$; r = $\sigma^{observer}$ / $\sigma^{trial}$; r = $\sigma^{sess(lab)}$/ $\sigma^{trial}$). Kinematic parameters for ankle-, knee- and hip-joints are presented by peak values (amplitude) and associated time point during the gait cycle.

| | | Parameter | Intraobserver | | Interobserver | | Between-Laboratory Intraobserver | |
|---|---|---|---|---|---|---|---|---|
| | | | $\sigma^{observer}$ | r | $\sigma^{repeated}$ | r | $\sigma^{sess(lab)}$ | r |
| Joint Angles [degree] | Hip | *Flexion ST* | 1.2 | 1.0 | 1.3 | 1.0 | 1.6 | 1.4 |
| | | *Extension ST* | 0.8 | 1.0 | 0.9 | 1.0 | 1.6 | 1.7 |
| | | *Flexion SW* | 0.7 | 1.0 | 0.9 | 1.0 | 1.2 | 1.5 |
| | | *Adduction ST* | 1.0 | 1.0 | 1.2 | 1.0 | 1.5 | 1.6 |
| | | *Abduction SW* | 0.7 | 1.0 | 0.9 | 1.2 | 1.3 | 1.4 |
| | Knee | *Flexion ST* | 2.1 | 1.0 | 2.4 | 1.0 | 2.3 | 1.2 |
| | | *Extension ST* | 1.3 | 1.0 | 1.9 | 1.3 | 1.7 | 1.4 |
| | | *Flexion SW* | 1.1 | 1.0 | 1.8 | 1.4 | 1.6 | 1.5 |
| | | *Range of motion* | 1.6 | 1.0 | 1.8 | 1.0 | 1.7 | 1.1 |
| | | *First abduction SW* | 0.8 | 1.0 | 2.4 | 3.2 | 4.2 | 5.1 |
| | | *Adduction SW* | 1.0 | 1.0 | 1.5 | 1.6 | 3.4 | 3.7 |
| | | *Second abduction SW* | 0.7 | 1.0 | 1.3 | 1.6 | 3.2 | 3.8 |
| | Ankle | *Plantarflexion ST* | 1.6 | 1.4 | 1.5 | 1.5 | 1.6 | 1.4 |
| | | *Dorsiflexion ST* | 1.7 | 1.3 | 1.5 | 1.3 | 1.4 | 1.1 |
| | | *Plantarflexion SW* | 2.5 | 1.1 | 3.8 | 1.7 | 3.6 | 1.5 |
| | | *Dorsiflexion SW* | 1.5 | 1.5 | 1.6 | 1.4 | 1.3 | 1.4 |
| Time at % Gait Cycle | Hip | *Flexion ST* | 1.4 | 1.0 | 1.9 | 1.2 | 1.3 | 1.2 |
| | | *Extension ST* | 0.8 | 1.0 | 0.9 | 1.2 | 1.2 | 1.2 |
| | | *Flexion SW* | 2.9 | 1.0 | 3.2 | 1.0 | 4.2 | 1.1 |
| | | *Adduction ST* | 3.8 | 1.0 | 4.4 | 1.0 | 5.4 | 1.0 |
| | | *Abduction SW* | 1.5 | 1.0 | 2.5 | 1.5 | 5.0 | 1.0 |
| | Knee | *Flexion ST* | 0.8 | 1.0 | 1.1 | 1.1 | 1.0 | 1.2 |
| | | *Extension ST* | 1.4 | 1.0 | 1.7 | 1.1 | 1.4 | 1.1 |
| | | *Flexion SW* | 0.9 | 1.0 | 1.0 | 1.1 | 0.9 | 1.2 |
| | | *First abduction SW* | 2.7 | 1.0 | 3.0 | 1.0 | 2.1 | 1.0 |
| | | *Adduction SW* | 2.0 | 1.0 | 3.1 | 1.0 | 2.4 | 0.9 |
| | | *Second abduction SW* | 1.7 | 1.0 | 3.0 | 1.2 | 2.3 | 0.9 |
| | Ankle | *Plantarflexion ST* | 1.0 | 1.0 | 1.3 | 1.4 | 1.1 | 1.3 |
| | | *Dorsiflexion ST* | 1.0 | 1.0 | 1.2 | 1.2 | 1.2 | 1.0 |
| | | *Plantarflexion SW* | 0.9 | 1.0 | 1.0 | 1.1 | 1.1 | 1.3 |
| | | *Dorsiflexion SW* | 4.2 | 1.0 | 4.4 | 1.1 | 4.8 | 1.1 |
| | **Analysis of Variance** | Intraobserver vs. Interobserver: $p < 0.001$; $\eta_p^2 = 0.498$ Intraobserver vs. Between-Laboratory Intraobserver: $p = 0.001$; $\eta_p^2 = 0.325$ Interobserver vs. Between-Laboratory Intraobserver: $p = 0.083$; $\eta_p^2 = 0.097$ | | | | | | |

**Remarks:** ST = stance phase; SW = swing phase.

### 3.2. Intraobserver Reliability

All intraobserver reliability variables fulfilled the assumption of normality. Intraobserver reliability ($ICC_{total}$ = 0.93) was significantly ($p < 0.001$, $\eta_p^2$ = 0.399) higher than interobserver reliability ($ICC_{total}$ = 0.79). When considering the total intraobserver ICC, excellent values were observed for 94% (29/31) of parameters, 3% (1/31) were fair to good and 3% (1/31) were poor (Table 1). Based on the lower limit of the 95% CI, 87% (27/31) of the parameters showed an ICC value of at least 0.75 (excellent reliability). For intraobserver reliability, the calculated standard errors ($\sigma^{repeated}$) and SEM were smaller than those for interobserver reliability ($\sigma^{repeated}$) (Table 3, Tabel A2); however, these values were not compared statistically. The highest error for intraobserver reliability existed for ankle dorsiflexion during the swing phase (% gait cycle) ($\sigma^{repeated}$ = 4.2%), as well as plantarflexion during the swing phase (mean $\sigma^{observer}$ = 2.5°).

Intraobserver reliability across laboratories (between-laboratory intraobserver) was excellent in 29% (9/31) of all parameters, fair to good for 48% (15/31), and poor for 26% (8/31) (Table 1). The highest ICC (0.90) was observed for hip flexion during the stance phase (% gait cycle). The worst ICC (0) was calculated for knee abduction angle in the swing phase.

### 3.3. Between-Laboratory Intraobserver Reliability

All between-laboratory intraobserver reliability variables fulfilled the assumption of normality. Between-laboratory intraobserver reliability for the entire lower limb (hip, knee, ankle) was poor ($ICC_{total}$ = 0.56, Table 2). Excellent ICC values were only calculated for 29% (9/31) of parameters (Table 1). Considering the lower limit of the 95% CI, no parameter reached an ICC level of 0.75. Calculated standard errors between-laboratory intraobserver reliability ($\sigma^{sess(lab)}$) were larger than those for interobserver and intraobserver reliability (Table 3). The largest standard errors were the same as those found for interobserver reliability, which included hip adduction (% gait cycle) at the stance phase with a mean $\sigma^{sess(lab)}$ = 5.4%, and for ankle dorsiflexion (% gait cycle) at the stance phase with a mean $\sigma^{sess(lab)}$ = 4.8%. The largest ratio was again observed for knee abduction in the swing phase ($\sigma^{sess(lab)}$ = 4.2°, r = 5.1) whereas its smallest inter-trial error was 0.8°.

### 3.4. Variance Analysis

When comparing the between-site CGA data from GL1 and GL2, the general linear model revealed significant differences (adjusted significance level: $p \leq 0.002$) for 39% of all parameters (12/31) (Table 4). Only three parameters (hip abduction swing phase (degree), hip extension stance phase (% gait cycle), knee flexion swing phase (% gait cycle)) fulfilled all three statistical criteria ($p \leq 0.002$ and $\eta_p^2 \geq 0.14$ and d $\geq 0.80$).

**Table 4.** Single observer comparison between laboratories (means, standard deviations, general linear model (GLM), standard error of measurement (SEM)). Significance level was set at 0.002. Large effect sizes (d ≥ 0.8; $\eta_p^2 \geq 0.14$) marked in bold.

| Parameter | GL1 | | GL2 | | Analysis of Variance | | | d | SEM |
|---|---|---|---|---|---|---|---|---|---|
| | Mean | SD | Mean | SD | F | *p* | $\eta_p^2$ | | |
| **Hip (degree)** | | | | | | | | | |
| *Flexion ST* | 36.5 | 2.79 | 38.1 | 1.68 | 2.44 | 0.121 | 0.02 | 0.72 | 1.41 |
| *Extension ST* | −7.19 | 2.00 | −6.57 | 3.67 | 20.0 | **<0.001** | **0.15** | 0.22 | 1.50 |
| *Flexion SW* | 35.9 | 2.34 | 37.6 | 1.58 | 0.03 | 0.855 | 0.00 | **0.87** | 1.06 |
| *Adduction ST* | 5.04 | 3.94 | 6.71 | 3.72 | 3.32 | 0.071 | 0.03 | 0.44 | 1.33 |
| *Abduction SW* | −8.44 | 2.12 | −7.02 | 1.37 | 141 | **<0.001** | **0.55** | **0.81** | 1.12 |
| **Hip (% gait cycle)** | | | | | | | | | |
| *Flexion ST* | 3.00 | 3.65 | 3.45 | 3.17 | 79.2 | **<0.001** | **0.40** | 0.13 | 1.08 |
| *Extension ST* | 52.0 | 1.64 | 50.7 | 1.10 | 35.2 | **<0.001** | **0.32** | **0.95** | 1.01 |
| *Flexion SW* | 88.7 | 5.32 | 86.6 | 3.94 | 0.13 | 0.721 | 0.00 | 0.45 | 3.85 |
| *Adduction ST* | 27.1 | 10.4 | 25.2 | 8.15 | 25.7 | **<0.001** | **0.18** | 0.21 | 4.35 |
| *Abduction SW* | 67.2 | 8.23 | 65.4 | 3.20 | 0.67 | 0.416 | 0.01 | 0.32 | 3.92 |
| **Knee (degree)** | | | | | | | | | |
| *Flexion ST* | 24.7 | 3.30 | 27.1 | 3.02 | 1.62 | 0.206 | 0.01 | 0.76 | 2.07 |
| *Extension ST* | 7.25 | 3.81 | 7.96 | 2.35 | 40.4 | **<0.001** | **0.26** | 0.23 | 1.57 |
| *Flexion SW* | 67.9 | 4.49 | 68.3 | 2.36 | 16.0 | **<0.001** | 0.12 | 0.12 | 1.45 |
| *Range of motion* | 60.7 | 1.72 | 60.4 | 2.27 | 3.35 | 0.070 | 0.03 | 0.15 | 1.47 |
| *First abduction SW* | 6.33 | 6.77 | 8.62 | 4.06 | 1.90 | 0.171 | 0.02 | 0.42 | 5.42 |
| *Adduction SW* | −0.07 | 7.73 | 0.46 | 2.48 | 9.26 | 0.003 | 0.08 | 0.10 | 3.92 |
| *Second abduction SW* | 4.96 | 7.33 | 5.83 | 1.60 | 0.00 | 0.960 | 0.00 | 0.20 | 3.43 |
| **Knee (% gait cycle)** | | | | | | | | | |
| *Flexion ST* | 12.9 | 1.70 | 11.8 | 1.82 | 12.1 | **0.001** | 0.09 | 0.63 | 0.77 |
| *Extension ST* | 39.1 | 1.49 | 38.7 | 1.72 | 0.09 | 0.769 | 0.00 | 0.25 | 1.43 |
| *Flexion SW* | 72.9 | 0.78 | 72.0 | 1.03 | 19.2 | **<0.001** | **0.14** | **1.00** | 0.84 |
| *First abduction SW* | 64.5 | 1.80 | 64.5 | 3.01 | 0.11 | 0.740 | 0.00 | 0 | 1.73 |
| *Adduction SW* | 75.1 | 2.21 | 76.2 | 2.24 | 0.80 | 0.374 | 0.01 | 0.49 | 2.00 |
| *Second abduction SW* | 85.3 | 3.52 | 85.7 | 3.37 | 4.45 | 0.035 | 0.04 | 0.12 | 1.72 |
| **Ankle (degree)** | | | | | | | | | |
| *Plantarflexion ST* | 0.06 | 2.79 | 1.34 | 1.11 | 39.1 | **<0.001** | **0.25** | 0.66 | 1.60 |
| *Dorsiflexion ST* | 18.0 | 2.66 | 17.5 | 2.00 | 3.06 | 0.083 | 0.03 | 0.22 | 1.19 |
| *Plantarflexion SW* | −11.3 | 10.7 | -7.68 | 6.69 | 110 | **<0.001** | **0.49** | 0.42 | 3.25 |
| *Dorsiflexion SW* | 8.69 | 2.96 | 9.76 | 2.77 | 9.06 | 0.003 | 0.07 | 0.37 | 1.18 |
| **Ankle (% gait cycle)** | | | | | | | | | |
| *Plantarflexion ST* | 6.43 | 1.83 | 5.22 | 1.65 | 12.7 | **0.001** | 0.10 | 0.70 | 0.84 |
| *Dorsiflexion ST* | 47.3 | 1.48 | 46.9 | 1.27 | 0.01 | 0.940 | 0.00 | 0.29 | 1.10 |
| *Plantarflexion SW* | 64.4 | 1.02 | 63.1 | 1.21 | 3.66 | 0.058 | 0.03 | **1.17** | 0.95 |
| *Dorsiflexion SW* | 91.4 | 5.37 | 87.7 | 5.37 | 0.20 | 0.656 | 0.00 | 0.69 | 3.98 |

**Remarks:** ST = stance phase; SW = swing phase; GL1 = laboratory 1; GL2 = laboratory 2.

### 3.5. Scatter-Plot Technique Suggested by Bland and Altman [22]

The Bland and Altman [22] scatter-plot technique revealed that the largest (worst) bounding criterion was for frontal plane knee angle (−2.3 ± 16.1°) (Figure 3d). The hip joint angles (Figure 3a,b), as well as the sagittal plane angles of the knee joint (Figure 3c,d), had the lowest bounding range. Ankle angles in general had a small bounding range (Figure 3e,f), with the exception of the maximum plantarflexion angle during the swing phase (−3.6 ± 10.7°).

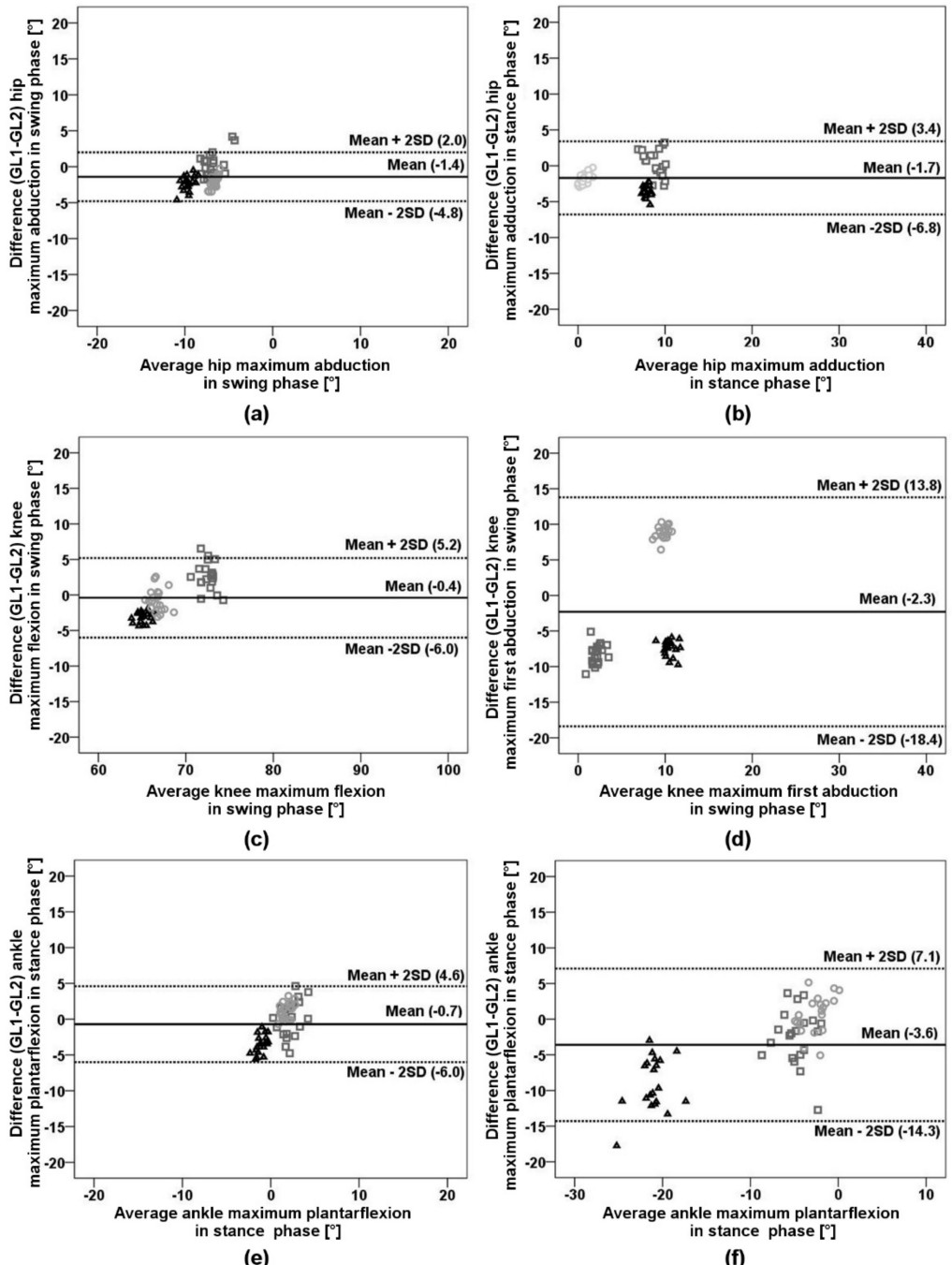

**Figure 3.** Bland and Altman plots presenting the computed differences between the clinical gait analysis data from the first laboratory (GL1) and the second laboratory (GL2) for (**a**,**b**) hip joint, (**c**,**d**) knee joint and (**e**,**f**) ankle joint. Each data point (test subject 1 = circle, test subject 2 = triangle, test subject 3 = square) represents the computed difference between CGA data of GL1 and GL2 (ordinate) which was plotted vs. the mean difference (abscissa) respectively, from the same healthy test subjects. The solid horizontal line represents the mean of all the differences plotted, and the two dashed lines represent the mean ± 2 SD (standard deviation) agreement interval, as defined by Bland and Altman [22].

### 3.6. Joint-Related Reliability

Inter- and intraobserver reliability for the ankle was in the fair-to-good and excellent ranges, in contrast to the knee and hip, which showed excellent reliability only (Table 2, Table A2). The within-site interobserver reliability for the ankle was $ICC_{mean} = 0.58$ (95% CI: 0.33–0.73) compared

with $ICC_{mean}$ = 0.81 (95% CI: 0.66–0.88) for intraobserver reliability. This was much lower than the reliability for the knee (interobserver $ICC_{mean}$ = 0.83 (95% CI: 0.74–0.89), intraobserver $ICC_{mean}$ = 0.98 (95% CI: 0.94–0.99)) and hip (interobserver $ICC_{mean}$ = 0.90 (95% CI: 0.84–0.94), intraobserver $ICC_{mean}$ = 0.97 (95% CI: 0.95–0.99)) (Table 2).

Between-laboratory intraobserver reliability showed fair-to-good reliability for all joints. Within this reliability level, small differences were found for the knee ($ICC_{mean}$ = 0.46 (95% CI: 0.05–0.68)) in comparison to the ankle (between-laboratory intraobserver $ICC_{mean}$ = 0.58) and hip (between-laboratory intraobserver $ICC_{mean}$ = 0.65 (95% CI: 0.27–0.81)).

Based on the $ICC_{total}$, significant differences were calculated between all three types of reliability ($p < 0.001$) (Table 2).

## 4. Discussion

The ability to exchange and compare clinical gait analysis data between laboratories is a valuable tool to improve the monitoring of clinical treatments and rehabilitation progressions for patients with musculoskeletal disease. Gait analysis data was thus compared in three ways in the present study. First, to determine interobserver reliability; second, to determine intraobserver reliability; and finally, to determine between-laboratory intraobserver reliability of CGA data between two gait laboratories.

In support of our first hypothesis, the total mean reliability was excellent for interobserver ($ICC_{total}$ = 0.79, 95% CI: 0.67–0.86) and intraobserver reliability ($ICC_{total}$ = 0.93, 95% CI: 0.87–0.96) (Table 2). For a single laboratory, we found fair-to-good or poor interobserver reliability, which suggests an observer influence dependent on subjective observer variability and the different software used for analysis (Workstation vs. Nexus) (Table 1). These results support those of previous work by Schwartz et al. [10] (Table 3). We found excellent intraobserver reliability for one observer processing the data using the same software (excluding observer and software error). Nonetheless, ICC values for inter- and intraobserver reliability still fall under the excellent classification for a majority of the parameters observed (interobserver: 71% parameters; intraobserver: 94% parameters).

The ankle displayed fair-to-good and excellent interobserver reliability, whereas the knee and hip showed excellent reliability only (Table 2). This outcome is in general agreement with the systematic review of McGinley et al. [3], who reported excellent intra- and interobserver reliability for the parameters of the hip and knee joints in the sagittal-plane, and a lower reliability for the ankle joint. However, the observed differences within ankle kinematics may in part be attributed to technical improvements in the applied software (Nexus, version 1.3 vs. Workstation, version 4.6 build 142). However, future research is needed to investigate this possibility.

For total mean reliability, we observed fair-to-good between-laboratory intraobserver reliability ($ICC_{total}$ = 0.56, 95% CI: 0.16–0.74) and excellent inter- ($ICC_{total}$ = 0.79, 95% CI: 0.67–0.86) and intraobserver reliability ($ICC_{total}$ = 0.93, 95% CI: 0.87–0.96). Only 26% of the between-laboratory intraobserver parameters revealed excellent ICC values, whereas the remaining 74% fell within the fair-to-good and poor ranges. Significant differences ($p \leq 0.002$) were found in 39% of the compared parameters (Table 4). This fair-to-good and poor reliability may be caused by differences in the measurement system hardware configurations and by slight differences in marker placement between data collection sessions. Both laboratories employed electro-optical motion capturing from the same manufacturer, but use a different hardware configuration (capturing software, camera type).

According to our second hypothesis, we believed that CGA data could be accurately collected between two gait laboratories, making such data interchangeable. An acceptance of this hypothesis could be affirmed for 61% of the parameters, with significant differences ($p \leq 0.002$) in the remaining 39% (Table 4). Between-laboratory intraobserver reliability of CGA data captured at two different gait analysis laboratories had total ICC values at the fair-to-good level, which was lower than the excellent inter- and intraobserver reliability values (Table 2). In general, differences were found at each joint (ankle, knee, hip) for angle as well as for time-dependent (% gait cycle) analysis. The Bland–Altman plots indicated a detectable amount of variation for specific parameters, such as the knee in the

frontal plane (Figure 3d), when compared between laboratories. These plots also showed a small data distribution for each individual observer, when compared between laboratories and the largest (worst) bounding criteria for frontal plane knee angle (2.3 ± 16.1°) (Figure 3d).

The standard errors for between-laboratory intraobserver analysis (maximum abduction at stance phase $\sigma^{sess(lab)}$ = 4.2, maximum adduction at swing phase $\sigma^{sess(lab)}$ = 3.4, maximum second abduction swing phase $\sigma^{sess(lab)}$ = 3.2) supports the low between-laboratory intraobserver reliability found for the knee ($ICC_{mean}$ = 0.46, Table 2). These values are twice as large as for inter- (maximum first abduction stance phase $\sigma^{observer}$ = 2.4; maximum adduction/abduction swing phase $\sigma^{observer}$ = 1.5; maximum second abduction swing phase $\sigma^{observer}$ = 1.3) and intraobserver reliability (maximum first abduction stance phase $\sigma^{repeated}$ = 0.8; maximum adduction/abduction swing phase $\sigma^{repeated}$ = 1.0; maximum second abduction swing phase $\sigma^{repeated}$ = 0.7). The lower between-laboratory intraobserver reliability and significant differences observed indicate an influence of the applied motion capturing camera technology on the captured CGA data between testing sites. In contrast to our measure of intra- and interobserver variability in the between-laboratory comparison, the effect of different hardware (cameras, lenses, Analog-to-digital (A/D) converter, etc.), as well as marker removal and replacement between data sets can also contribute to lower reliability. The reduction in reliability suggests that the detection of tracked markers may be affected by the different camera resolutions investigated (Vcam with 0.3 million pixels, 659 x 439 black/white pixel sensor resolution vs. MXF with 2 million pixels, 1600 x 1280 grayscale pixel sensor resolution). We believe the effect of the measurement protocol and observer dependence are probably small in comparison to available technical requirements, existing in the different generations of cameras (i.e., sensor technology) because the ICC values of the one-site interobserver reliability were much higher, falling well within the excellent classification. These findings were somewhat different than those reported by Bucknall et al. [11], who observed apparent differences during motion data captured simultaneously using three camera systems (612, MX-13 and MX-F40).

Results of this study support the findings of Gorton et al. [8] and Bucknall et al. [11], which showed a dependence of CGA data quality on system resources, as well as applying standardized measurement protocols. The fact that CGA data quality could be affected by different camera types, even ones from the same manufacturer, is problematic, not only in the context of multicenter investigations, but it could also be a problem when comparing CGA data within institutions that have recently updated their laboratory equipment. Based on these findings when multicenter investigations are considered, it is important that both laboratories use the same or comparable camera equipment and software.

A limitation of this investigation was the sample size (*n* = 3). Each subject performed 20 barefoot gait trials using a self-selected walking speed. Subjects had no gait pathologies, resulting in similar gait patterns. This approach included the risk of correlated observations. On the other hand, it was not feasible to conduct our investigation with actual patients (with gait pathologies) and a powered sample, due to time restrictions in subject preparation (e.g., time required for marker application) and transportation limitations between testing sites.

## 5. Conclusions

The results of this study showed higher intraobserver reliability than interobserver reliability for CGA data conducted in a single gait analysis laboratory. There was weaker intra- and interobserver reliability ankle kinematics when compared to the hip and knee. Inter- or intraobserver reliability for data collected at a single laboratory was much stronger compared to data collected between laboratories. The outcomes of this study indicate that CGA results are probably influenced by testing conditions, such as laboratory equipment and software (capturing software and camera type). Peer-reviewed literature has reported an effect on the repeatability of motion capture data using separate trials, sessions and observers, etc. [5–7,9,10], and for multicenter repeatability focused on different applied hardware configurations, marker placement, as well as between trials and days of measurement [3,8,11]. The results of the current study support these previous investigations, and suggest that when

multicenter investigations are considered, it is important that both laboratories use the same or comparable camera equipment and software. The reduced between-laboratory intraobserver reliability implies that comparisons of CGA data between centers with varying measurement equipment are generally not recommended.

**Author Contributions:** Conceptualization, R.S., F.S., C.H. and R.W.; methodology, R.W., R.S., F.S. and C.H.; formal analysis, R.W., R.S. and F.S.; investigation, R.W., R.S. and F.S.; resources, C.H.; data curation, R.S.; writing—original draft preparation, R.W., R.S. and F.S.; writing—review and editing, R.S., F.S., C.H., K.L. and R.W.; visualization, F.S.; supervision, C.H.; project administration, R.S. and F.S.; All authors have read and agreed to the published version of the manuscript.

**Funding:** This research received no external funding.

**Acknowledgments:** The authors wish to thank Gavin D. Olender for his assistance during preparing this manuscript. We would although like to thank Martin Scott-Löhrer from prophysics AG (Zürich, Switzerland) for his assistance and support. We thank Siegfried Leuchte and Henning Windhagen for the supervision and support of this project. This manuscript is dedicated to Siegfried Leuchte, who passed away unexpectedly at the end of 2018. We honor him as a committed and insightful Professor and keep him in honorable memory. This research was made possible out of research cooperation between members of the Musculosceletal Biomechanics Network (www.msb-net.org) of the DGOOC (German Association of Orthopedics and Orthopedic Surgery).

**Conflicts of Interest:** The authors declare no conflict of interest.

## Appendix A

**Table A1.** Anthropometric data for each individual test subject.

|  | Age [years] | Body Weight [kg] | Body Height [cm] | Body Mass Index [kg·m$^{-2}$] |
|---|---|---|---|---|
| Subject 1 | 29 | 59 | 167 | 21 |
| Subject 2 | 37 | 83 | 178 | 26 |
| Subject 3 | 34 | 90 | 185 | 26 |
| Mean | 33.3 | 77.3 | 177 | 24.3 |
| SD | 4.04 | 16.3 | 9.07 | 2.89 |

**Table A2.** Descriptive statistics (mean ± standard deviation) and SEM values reported for intra- and interobserver reliability to describe the variability within and between subjects.

| Parameter | Intraobserver | | | Interobserver | | |
|---|---|---|---|---|---|---|
| | Exam 1 | Exam 2 | SEM | Observer 1 | Observer 2 | SEM |
| **Hip (degree)** | | | | | | |
| Flexion ST | 38.2 ± 1.73 | 38.1 ± 1.68 | 0 | 38.1 ± 1.68 | 38.1 ± 1.69 | 0.29 |
| Extension ST | −6.58 ± 3.68 | −6.57 ± 3.67 | 0 | −6.57 ± 3.67 | −6.24 ± 4.49 | 2.00 |
| Flexion SW | 37.6 ± 1.62 | 37.6 ± 1.58 | 0 | 37.6 ± 1.58 | 37.7 ± 1.61 | 0.16 |
| Adduction ST | 6.69 ± 3.71 | 6.71 ± 3.72 | 0 | 6.71 ± 3.72 | 6.53 ± 3.74 | 0.53 |
| Abduction SW | −7.01 ± 1.38 | −7.02 ± 1.37 | 0 | −7.02 ± 1.37 | −6.96 ± 1.45 | 0.37 |
| **Hip (% gait cycle)** | | | | | | |
| *Flexion ST* | 3.55 ± 3.19 | 3.45 ± 3.17 | 0 | 3.45 ± 3.17 | 3.73 ± 3.38 | 0.66 |
| *Extension ST* | 51.0 ± 1.17 | 50.7 ± 1.10 | 0.32 | 50.7 ± 1.10 | 51.0 ± 1.13 | 0.49 |
| *Flexion SW* | 86.6 ± 3.05 | 86.6 ± 3.94 | 1.44 | 86.6 ± 3.94 | 86.2 ± 3.26 | 1.94 |
| *Adduction ST* | 25.2 ± 8.38 | 25.2 ± 8.15 | 0 | 25.2 ± 8.15 | 24.8 ± 7.81 | 1.78 |
| *Abduction SW* | 65.7 ± 3.39 | 65.4 ± 3.20 | 0.33 | 65.4 ± 3.20 | 65.1 ± 3.53 | 0.82 |
| **Knee (degree)** | | | | | | |
| *Flexion ST* | 27.2 ± 3.03 | 27.1 ± 3.02 | 0 | 27.1 ± 3.02 | 27.1 ± 3.16 | 0.93 |
| *Extension ST* | 7.91 ± 2.39 | 7.96 ± 2.35 | 0 | 7.96 ± 2.35 | 7.86 ± 2.44 | 0 |
| *Flexion SW* | 68.3 ± 2.33 | 68.3 ± 2.36 | 0 | 68.3 ± 2.36 | 67.7 ± 4.58 | 2.53 |
| *Range of motion* | 60.4 ± 2.27 | 60.4 ± 2.27 | 0 | 60.4 ± 2.27 | 59.9 ± 4.48 | 2.61 |
| *First abduction SW* | 8.61 ± 4.03 | 8.62 ± 4.06 | 0 | 8.62 ± 4.06 | 8.44 ± 3.81 | 0.39 |
| *Adduction SW* | 0.44 ± 2.41 | 0.39 ± 2.45 | 0 | 0.39 ± 2.45 | 0.60 ± 2.33 | 0.41 |
| *Second abduction SW* | 5.79 ± 1.58 | 5.86 ± 1.60 | 0 | 5.86 ± 1.60 | 5.75 ± 1.47 | 0.27 |
| **Knee (% gait cycle)** | | | | | | |
| *Flexion ST* | 11.9 ± 1.88 | 11.8 ± 1.82 | 0.19 | 11.8 ± 1.82 | 11.9 ± 1.69 | 0.61 |
| *Extension ST* | 38.8 ± 1.75 | 38.7 ± 1.72 | 0.25 | 38.7 ± 1.72 | 38.4 ± 1.89 | 0.44 |
| *Flexion SW* | 72.4 ± 0.97 | 72.0 ± 1.03 | 0.33 | 72.0 ± 1.03 | 71.9 ± 1.09 | 0.42 |
| *First abduction SW* | 65.0 ± 3.22 | 64.5 ± 3.01 | 0.54 | 64.5 ± 3.01 | 64.8 ± 2.96 | 1.19 |
| *Adduction SW* | 76.6 ± 2.20 | 76.1 ± 2.23 | 0.31 | 76.1 ± 2.23 | 75.5 ± 3.11 | 1.41 |
| *Second abduction SW* | 86.4 ± 3.31 | 85.8 ± 3.30 | 0.47 | 85.8 ± 3.30 | 85.8 ± 3.78 | 1.06 |
| **Ankle (degree)** | | | | | | |
| *Plantarflexion ST* | 0.37 ± 1.32 | 1.34 ± 1.11 | 1.07 | 1.34 ± 1.11 | −0.17 ± 1.24 | 1.11 |
| *Dorsiflexion ST* | 16.6 ± 1.11 | 17.5 ± 2.00 | 0.97 | 17.5 ± 2.00 | 16.0 ± 1.68 | 1.34 |
| *Plantarflexion SW* | −8.66 ± 6.08 | −7.68 ± 6.69 | 0.90 | −7.68 ± 6.69 | −9.01 ± 5.64 | 1.63 |
| *Dorsiflexion SW* | 8.92 ± 1.56 | 9.76 ± 2.77 | 0.92 | 9.76 ± 2.77 | 8.45 ± 2.25 | 1.33 |
| **Ankle (% gait cycle)** | | | | | | |
| *Plantarflexion ST* | 5.35 ± 1.49 | 5.22 ± 1.65 | 0.22 | 5.22 ± 1.65 | 5.57 ± 1.31 | 0.42 |
| *Dorsiflexion ST* | 47.0 ± 1.41 | 46.9 ± 1.27 | 0.30 | 46.9 ± 1.27 | 45.7 ± 4.46 | 2.53 |
| *Plantarflexion SW* | 63.4 ± 1.12 | 63.1 ± 1.21 | 0.33 | 63.1 ± 1.21 | 62.9 ± 1.04 | 0.45 |
| *Dorsiflexion SW* | 87.6 ± 5.16 | 87.7 ± 5.37 | 1.18 | 87.7 ± 5.37 | 85.7 ± 3.81 | 3.25 |

**Remarks:** ST = stance phase; SW = swing phase; Exam = examination.

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
