# Peer review of "Intra- and Interobserver Reliability Comparison of Clinical Gait Analysis Data between Two Gait Laboratories"

_applsci, doi:10.3390/app10155068_

Round 1

Reviewer 1 Report

Review of Applied Sciences manuscript #875749, “Intra- and interobserver reliability of clinical gait analysis data within a multicenter trial”

General Comments:

This manuscript describes a study to evaluate the reliability of gait analysis performed at two locations using different software and hardware and when data were evaluated by multiple analysts.

In general, the study seems well performed and the manuscript is well written and adequately describes the study. There are a few relatively minor concerns with the manuscript as written.

Specific comments:

Introduction

Line 87, please define “single data pool” and then in line 90, manuscript refers to “same cohort of subjects”, so reader is left to wonder, what is difference between “single data pool” and “same cohort of subjects” ? are the same or different ? and if same, why not use one term ?

Methods

Lines 142-144: Why is this information in methods section ? Isn’t this information a result of the study and belongs in results section ? “No parameters differed 142 between laboratory measurement sessions (walking speed: 1.40 ± 0.06 m/s vs. 1.40 ± 0.04 m/s; stride 143 length: 1.38 ± 0.09 m vs. 1.40 ± 0.10 m).”

Line 194: Wait, if “two standard deviations”, then shouldn’t this be 95% of all measured values and not 98% ?? “two standard deviations of the measured differences (approximately 98% of all measured 194 values)”.

Reviewer 2 Report

This study investigated the intra- and interobserver reliability of multiple gait analysis variables obtained from an optoelectronic camera-based system. The study included 3 healthy young participants presented with 20 trials of normal paced walking in a laboratory setting. The reliability estimates were calculated for: 1) inter-observer (same dataset collected by the same person, same site) but 2 different analysis software used by 2 different; 2) intra-observer within laboratories (same dataset, same software, same site) data reanalysed with 12 months later; 3) Intra-observer between laboratories (same observer, same dataset, same software).

The manuscript is well written and analysis are sound and clinically relevant. Introduction is relevant and posts the problem well, although it seems that there is some previous research already done in the topic. Here the authors should highlight more the uniqueness and relevance of this study – what new information it can provide.

The authors have made a good effort to describe the analyses and comparisons made as well as possible. Still, the multi-center term is confusing since there were only 2 sites. Suggestion: Change the “Multicenter reliability” to Between-laboratory inta-observer reliability. No need to use multi when there were only 2 sites; be clear and say between-laboratory, or between 2 laboratories.

Likewise, the title should be changed accordingly: e.g. ... within two laboratories OR between laboratories or similar. Please change multi-centre everywhere in the manuscript text.

Intra-observer reliability analysis could also be done for GL1, that would give valuable information in case these differ.

It is not very clear how the 20 trials from 3 participants were handled in reliability analysis. Please add some explanation in the manuscript.

How was the within and between subject variability handled in the analysis OR was it totally ignored. This needs to be mentioned in the discussion (Limitations)

No reference is provided for interpretation of ICC. Commonly, 0.75 is defined as good, and 0.90 as excellent; 0.40 could be fair, and less than 0.40 poor. Possible 0.75 could be interpreted as good-to-excellent if you put those ranged together, but 0.40-0.75 would still be fair.

The Bland-Altman plots were very good, gives possibility to see data separated between and within subjects.

I would like to see more elaborated discussion around limitations of this study, e.g. how the correlated data (20 trials) influence the ICC values. In addition, the 3 subjects seem to be relatively similar to each other in age and maybe other characteristics.

Similarly to Introduction, it would be valuable to highlight what unique information can this study provide to the research area in large or specifically.

Minor comments:

Please provide individual background data on 3 participants and not only means and SD. (possibly as supplement)

Statistics, point 4: confusing with different terms used in the equation 2 (observer and trials) compared to defined errors listed above.

I can’t see that the analysis of variance is described in the statistics section.
